

# A tyrannosauroid metatarsus from the Merchantville Formation of Delaware increases the diversity of non-tyrannosaurid tyrannosauroids on Appalachia

Chase D. Brownstein

Collections and Exhibitions, Stamford Museum & Nature Center, Stamford, CT, USA

## ABSTRACT

During the Late Cretaceous, the continent of North America was divided into two sections: Laramidia in the west and Appalachia in the east. Although the sediments of Appalachia recorded only a sparse fossil record of dinosaurs, the dinosaur faunas of this landmass were different in composition from those of Laramidia. Represented by at least two taxa (*Appalachiosaurus montgomeriensis* and *Dryptosaurus aquilunguis*), partial and fragmentary skeletons, and isolated bones, the non-tyrannosaurid tyrannosauroids of the landmass have attracted some attention. Unfortunately, these eastern tyrants are poorly known compared to their western contemporaries. Here, one specimen, the partial metatarsus of a tyrannosauroid from the Campanian Merchantville Formation of Delaware, is described in detail. The specimen can be distinguished from *A. montgomeriensis* and *D. aquilunguis* by several morphological features. As such, the specimen represents a potentially previously unrecognized taxon of tyrannosauroid from Appalachia, increasing the diversity of the clade on the landmass. Phylogenetic analysis and the morphology of the bones suggest the Merchantville specimen is a tyrannosauroid of "intermediate" grade, thus supporting the notion that Appalachia was a refugium for relict dinosaur clades.

## INTRODUCTION

The fossil record of Appalachia, a landmass formed from the eastern half of North America by the expansion of the Western Interior Seaway (*Russell, 1995*; *Roberts & Kirschbaum, 1995*; *Lehman, 1997*; *Schwimmer, 1997*, *2002*), is poor in contrast to that of Cretaceous-aged western North America (Laramidia), with dinosaurs only represented by isolated elements and rare partial skeletons (*Gallagher, 1993*; *Schwimmer, 1997*; *Ebersole & King, 2011*; *Gangloff, 2012*). Nevertheless, the dinosaur faunas of Appalachia have gradually come to light, consisting of hadrosauroids, hadrosaurids, nodosaurids, leptoceratopsians, indeterminate ornithopods, dromaeosaurids,

Corresponding author
Chase D. Brownstein,
chasethedinosaur@gmail.com

ornithomimosaurs, indeterminate maniraptorans, and non-tyrannosaurid tyrannosauroids (e.g., *Gallagher, 1993*; *Schwimmer, 1997*, *2002*; *Ebersole & King, 2011*; *Schwimmer et al., 2015*; *Longrich, 2016*). The tyrannosauroid genera of this landmass, *Dryptosaurus* and *Appalachiosaurus*, have been placed outside Tyrannosauridae in multiple phylogenetic analyses (*Holtz, 2004*; *Carr, Williamson & Schwimmer, 2005*; *Brusatte et al., 2010*; *Brusatte, Benson & Norell, 2011*; *Loewen et al., 2013*; *Fiorillo & Tykoski, 2014*; *Brusatte & Carr, 2016*; *Brusatte et al., 2016*; *Carr et al., 2017*). *Brusatte & Carr (2016)* also found *Dryptosaurus aquilunguis* as an alioramine tyrannosauroid in their Bayesian analysis, though this result was not corroborated by a more recent parsimony analysis of Tyrannosauroidea using the same dataset with additional taxa and added characters (*Carr et al., 2017*). The fossil record of tyrannosauroids from Appalachia is restricted to the holotypes of *Dryptosaurus* and *Appalachiosaurus*, undescribed partial specimens, and isolated bones and teeth, limiting our understanding of these animals. Overall, the scarcity of tyrannosauroid specimens from Appalachia means that the description of additional specimens and their inclusion in phylogenetic analysis is needed to better illuminate the understanding of the diversity, biogeography, and evolutionary position of eastern tyrant dinosaurs.

One productive area for Late Cretaceous dinosaur specimens in eastern North America is the portion of the Atlantic Coastal Plain that includes New Jersey and Delaware (e.g., *Gallagher, 1993*). Cretaceous sediments in this area range in age from the Cenomanian to the Maastrichtian stages of the Late Cretaceous (*Miller et al., 2004*). In this paper, the partial metatarsus of a tyrannosauroid from the early Campanian Merchantville formation of Delaware is described. The specimen, YPM VPPU.021795, can be distinguished from *Appalachiosaurus montgomeriensis* (*Carr, Williamson & Schwimmer, 2005*) and *D. aquilunguis* (*Cope, 1866*; *Marsh, 1877*; *Carpenter et al., 1997*; *Brusatte, Benson & Norell, 2011*) by a variety of morphological features and may represent a previously unrecognized taxon of tyrannosauroid from Appalachia, adding diversity to the clade in eastern North America. Both phylogenetic analysis and the presence of several features on YPM VPPU.021795 strongly support a position outside of Tyrannosauridae for the specimen, further evincing the presence of "intermediate" tyrannosauroids on Appalachia (*Carr, Williamson & Schwimmer, 2005*; *Brusatte, Benson & Norell, 2011*) and providing additional evidence that at least some Appalachian dinosaurs represent relict forms that survived on the landmass throughout the Late Cretaceous (*Schwimmer, 1997*).

## METHODS

### Permits

No permits were required for this study, which complied with all relevant regulations. Access to the collections at the American Museum of Natural History was provided by Carl Mehling, whereas access to the collections at the Yale Peabody Museum was provided by Daniel Brinkman.

## Photography

The specimens described herein were photographed using a Canon Powershot G-12 digital camera and cropped using Apple Preview.

## Phylogenetic analysis protocols

YPM VPPU.021795 was included in both the latest published version of the Theropod Working Group matrix (*Brusatte et al., 2014a*) (Data S1 and S2) to statistically test the placement of the specimen in Coelurosauria and the matrix of *Carr et al. (2017)* (Data S1 and S3) to better resolve the placement of YPM VPPU.021795 within Tyrannosauroidea. The matrices were entered into TNT v. 1.5 in the command.tnt line in an analysis of parsimony (*Goloboff & Catalano, 2016*). In order to make the analysis of YPM VPPU.021795 in the matrices of *Brusatte et al. (2014a)* and *Carr et al. (2017)* comparable to the protocols of those studies, the author followed the first's procedure of using *Allosaurus* and the second's of using *Allosaurus* sp., Maniraptora, Ornithomimosauria, and *Compsognathus longipes* as outgroup taxa. Following *Carr et al. (2017)*, *Raptorex* was regarded as a juvenile tyrannosaurine and excluded from the analysis and *A. altai* and *A. remotus* were entered into the program as one taxon. The "New Technology Search" was used in both analyses with default parameters for the options of sectorial search, ratchet, tree drift, and tree fuse. For the analysis of YPM VPPU.021795 within the matrix of *Carr et al. (2017)*, 32 most parsimonious trees (MPTs) were recovered, whereas in the analysis in the Theropod Working Group matrix, 81 were found. These trees were then subjected to traditional TBR branch swapping, which recovered no additional MPTs. Clade support was quantified by use of bootstrap values (100 replicates; *Goloboff & Catalano, 2016*).

# RESULTS

## Geological setting

YPM VPPU.021795 is an associated specimen collected by Ralph Johnson and Ray Meyer of the Monmouth Amateur Paleontologist's Society from a single outcrop of the Merchantville Formation exposed at low tide along a portion of the southern bank of the Chesapeake and Delaware Canal north of Summit, Delaware. Both metatarsals are reddish-brown in color, similar in length (Table 1), from the left pes of a theropod, of similar preservation (Figs. 1 and 2), individually assignable to tyrannosauroids, and found associated in the same deposit, indicating they belong to the same individual.

*Gallagher (1993)* described the Merchantville Formation as consisting of black to dark greenish-gray micaceous, silty clay to fine sand, noting its early Campanian age. *Miller et al. (2004)* found the Merchantville Formation to be latest Santonian to early Campanian in age (84.3–77.8 Ma), seated within the Merchantville sequence, and comprised of glauconite sands and glauconitic clays. In addition to the tyrannosauroid YPM VPPU.021795, tetrapods known from the Merchantville Formation include crocodylians, the mosasaurs Clidastes iguanavus and Halisaurus platyspondylus, the

**Table 1 Measurements of YPM VPPU.021795.**

| Measurement | Metatarsal II | Metatarsal IV |
|---|---|---|
| Proximodistal length (lateral) | 310 mm (est. 465 mm) | 312 mm (proximal), 75 mm (distal) (total 387 mm) (est. 464 mm) |
| Dorsoventral width (prox. end) | 80 mm | 63 mm |
| Mediolateral width (prox. end) | 55 mm (at the articular facet) | 48 mm (at the articular facet) |
| Dorsoventral width (midshaft) | 42 mm | 36 mm |
| Mediolateral width (midshaft) | 27 mm | 28 mm |
| Circumference (midshaft) | 112+ mm (crushed) | 131 mm |
| Dorsoventral width (dist. hemicondyles) | N/A | 36 mm |
| Mediolateral width (dist. hemicondyles) | N/A | 27 mm |
| Robusticity index (circumference at midshaft/total length) | N/A | 0.2823 |

turtles Bothremys cooki and Trionyx halophila, the dinosaur Hadrosaurus foulkii, and indeterminate hadrosaurid remains (*Gallagher, 1993*; *Weishampel & Young, 1996*).

At the site of discovery, the Chesapeake and Delaware Canal, *Baird & Galton (1981)* described the Merchantville Formation as a deposit consisting of silty and clayey glauconitic and micaceous thick-bedded sand approximately 23 feet in thickness, describing pterosaur specimens from the site and noting the presence of fossils of the turtle *Toxochelys*, the mosasaur *Tylosaurus*, and hadrosaurid dinosaur remains nearby.

## Systematic paleontology

Dinosauria *Owen (1842)* sensu *Padian & May (1993)*

Theropoda *Marsh (1881)* sensu *Gauthier (1986)*

Coelurosauria *von Huene (1914)* sensu *Sereno, McAllister & Brusatte (2005)*

Tyrannosauroidea *Osborn (1905)* sensu *Holtz (2004)*

Tyrannosauroidea indet.

**Material:** YPM VPPU.021795, partial left metatarsals II and IV of a tyrannosauroid dinosaur.

**Referral to Tyrannosauroidea:** The specimen YPM VPPU.021795 is referred to a tyrannosauroid based on several features that unite the specimen with that clade and larger, more inclusive ones within Theropoda. The presence of buttressing surfaces and proximal articular facets on each of the two preserved metatarsals (Figs. 1 and 2) unite YPM VPPU.021795 with theropod dinosaur groups with the arctometatarsalian condition (*Holtz, 1994*). In YPM VPPU.021795, the proximal articular surface of metatarsal IV appears as a deep, V-shaped notch on the medial surface of the bone that articulated with the lateral surface of the proximal end of a crescentic metatarsal III to form an arctometatarsus where the metatarsals were closely appressed (Figs. 2B and 2E), a morphology autapomorphic to Tyrannosauroidea (e.g., *Holtz, 2004*; *Thomson, Irmis & Loewen, 2013*; character 423 in

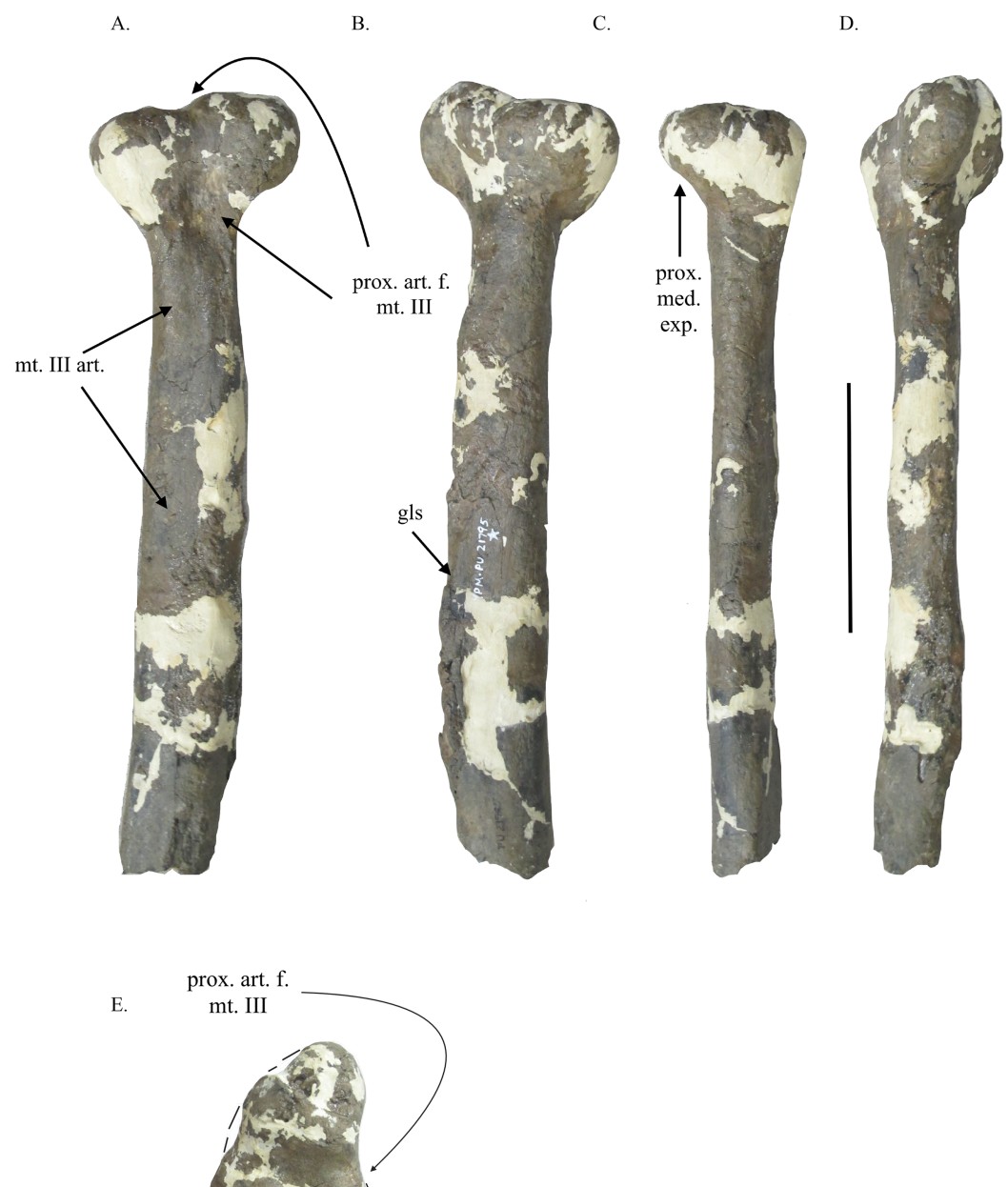

**Figure 1 Metatarsal II of YPM VPPU.021795.** Metatarsal IV in lateral (A), medial (B), dorsal (C), ventral (D), proximal (E), and distal (D) views. Abbreviations: mt. III art., articular surface for metatarsal III; prox. art. f. mt. III, proximal articular facet for metatarsal III; gls, M. gastrocnemius insertion scar; prox. med. exp., proximal medial expansion of metatarsal II. Scale bar = 100 mm.

*Brusatte et al., 2014a*; *Peecook et al., 2014*). The buttressing surfaces on the medial face of metatarsal IV and the lateral surface of metatarsal II of YPM VPPU.021795 are large and teardrop-shaped, another similarity with the condition in tyrannosauroids (e.g., character 303 in *Brusatte et al., 2010*; *Thomson, Irmis & Loewen, 2013*; *Peecook et al., 2014*; character

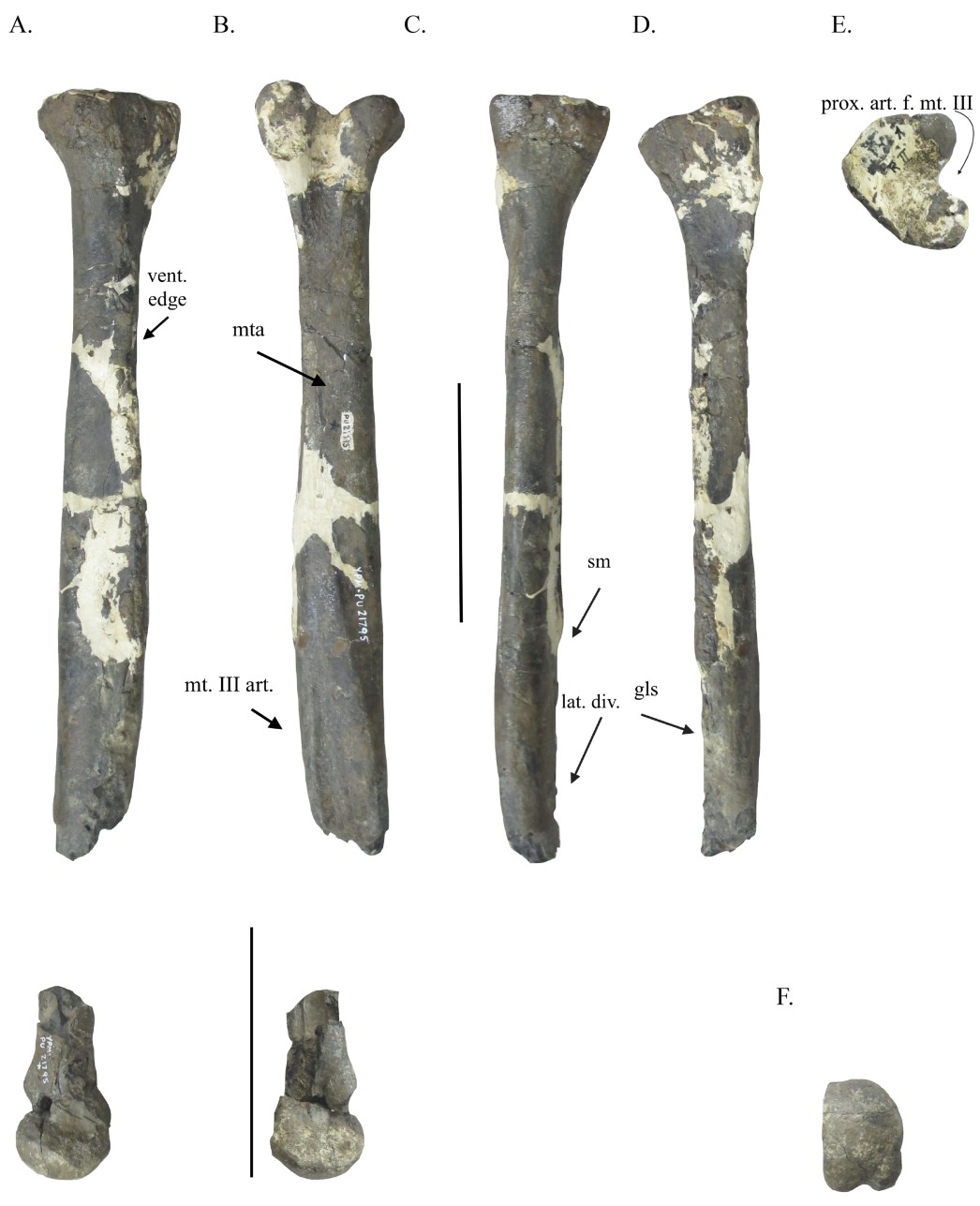

**Figure 2 Metatarsal IV of YPM VPPU.021795.** Metatarsal IV in lateral (A), medial (B), dorsal (C), ventral (D), proximal (E), and distal (D) views. Abbreviations: vent. edge, ventral edge; mta, M. tibialis interior insertion site; mt. III art., articular surface for metatarsal III; sm, shark feeding marks; lat. div., lateral divergence of metatarsal IV; gls, M. gastrocnemius lateralis insertion scar; prox. art. f. mt. III, proximal articular facet for metatarsal III. Scale bar = 100 mm.

303 in *Brusatte & Carr, 2016*). All of these features have previously been used to assign the isolated metatarsals and fragmentary hindlimbs of arctometatarsalian theropods to Tyrannosauroidea (*Thomson, Irmis & Loewen, 2013*; *Peecook et al., 2014*), so this assignment is not without methodical precedent. The presence of proximal articular surfaces for metatarsal III on metatarsals IV and II is also found in arctometatarsalian
ornithomimosaurs, caenagnathids, and troodontids (*Holtz, 1994*), and as such rejection of the assignment of YPM VPPU.021795 to these groups is discussed below.

Ornithomimids share several features in their metatarsals with tyrannosauroids, and as *Baird & Galton (1981)* assigned YPM VPPU.021795 to an ornithomimosaur, explicit rejection of the hypothesis that YPM VPPU.021975 belongs to an ornithomimosaur is especially warranted. In arctometatarsalian ornithomimosaurs, the proximal articular facets on metatarsals II and IV for metatarsal III are poorly developed and shallow to accommodate a proximally ovoid metatarsal III (*Osborn, 1921*; *Osmólska, Roniewicz & Barsbold, 1972*; *Barsbold & Osmólska, 1990*; *Shapiro et al., 2003*; *Makovicky, Kobayashi & Currie, 2004*; *Xu et al., 2011*; *Brusatte et al., 2012*; *Cullen et al., 2013*; *Peecook et al., 2014*; *McFeeters et al., 2016*; *Sues & Averianov, 2016*; *Tsogtbaatar et al., 2017*). This morphology is not seen in YPM VPPU.021795, where the proximal articular facet of metatarsal IV shows the bone articulated with the proximally crescentic metatarsal III autapomorphic for Tyrannosauroidea (Fig. 2E; *Holtz, 2004*). This indicated shape for metatarsal III also distinguishes YPM VPPU.021795 from the other arctometatarsalian groups noted above, and the size of the bones further restrict them referral to Troodontidae (*Makovicky & Norell, 2004*; *Peecook et al., 2014*). The preserved bone surface of metatarsal IV in YPM VPPU.021795 indicates that the metatarsal IV was laterally divergent distally that further distinguishes the specimen from ornithomimosaurs and caenagnathids (e.g., character 302 in *Brusatte et al., 2010*; *Peecook et al., 2014*; character 302 in *Brusatte & Carr, 2016*). The lack of a deep proximal articular facet on the metatarsal II included in YPM VPPU. 021795 as preserved warrants rigorous comparison of this bone with the equivalent elements of arctometatarsalian ornithomimosaur genera where the condition is somewhat similar (*Makovicky, Kobayashi & Currie, 2004*; *Brusatte et al., 2012*). Indeed, several other features of the metatarsal II suggest the bone came from a tyrannosauroid. Factoring in the clear damage and deformation of the bone (Fig. 1), the presence of a deepened facet on the medial surface of metatarsal II between the proximal articular surface and the buttressing surface for metatarsal III suggests that a deeper proximal articular facet was once present (Figs. 1B, 1C and 1E). The presence of a teardrop-shaped buttressing surface for metatarsal III on this bone also allies it with Tyrannosauroidea (*Holtz, 2004*; *Thomson, Irmis & Loewen, 2013*; *Loewen et al., 2013*). The medial extension outward of the Merchantville metatarsal II proximally as indicated by the preserved bone (Fig. 2C) does not match the condition in arctometatarsalian ornithomimosaurs, because the medial divergence proximally of metatarsal II is not sharply divergent from the shaft if present at all in the metatarsi of that group (e.g., Fig. 3A1, C in *Osborn, 1921*; Fig. 16A1 in *Osmólska, Roniewicz & Barsbold, 1972*; Fig. 6A in *Xu et al., 2011*; Fig. 10F, J in *McFeeters et al., 2016*; Fig. 21B in *Sues & Averianov, 2016*). This morphology more closely matches the metatarsals II of tyrannosauroids (e.g., Fig. 44A in *Lambe, 1917*; Fig. 1 in *Currie, 1998*; Fig. 19C in *Carr, Williamson & Schwimmer, 2005*). The size and elongate morphology of the Merchantville metatarsal II also distinguish the bone from the metatarsals of the other known theropod taxa of Appalachia, including the proximodistally robust and non-arctometatarsalian metatarsals of dromaeosaurids (*Keirnan & Schwimmer, 2004*; *Norell & Makovicky, 2004*; *Ebersole & King, 2011*; *Schwimmer et al., 2015*). Finally, the association and comparability of

this bone with the metatarsal IV described herein that clearly bears the deep, sharply pointed notch for the articulation with a crescentic metatarsal III autapomorphic for Tyrannosauroidea (*Holtz, 2004*) warrant the assignment of the metatarsal II to that clade.

## Phylogenetic analysis

Previous studies on fragmentary or isolated tyrannosauroid postcranial remains have noted the lack of characters from the metatarsus in character lists for the phylogenetic analysis of Tyrannosauroidea (*Peecook et al., 2014*), with some studies on fragmentary hindlimb material not performing a phylogenetic analysis at all (*Thomson, Irmis & Loewen, 2013*; *Brusatte et al., 2014b*: Data deposited in the Dryad repository: https://doi.org/10.5061/dryad.84t75). Indeed, only five of the 386 characters (~1.3%) included in the character list of *Carr et al. (2017)* could be coded for using YPM VPPU.021795. The characters scored were 377[1], 380[1], 381[0], 383[1], and 384[0], and the matrix was analyzed using TNT v.1.5 (*Goloboff & Catalano, 2016*). The tyrannosauroid represented by YPM VPPU.021795 was found to be an intermediate-grade tyrannosauroid closely related to *Timurlengia*, *Appalachiosaurus*, *Dryptosaurus*, and the Iren Dabasu tyrannosauroid in each of the 32 MPTs found (tree length = 782, consistency index = 0.574, retention index = 0.49), the strict consensus placing the Merchantville specimen as an outgroup to the clade containing *Bistahieversor* and Tyrannosauridae along with *Dryptosaurus*, *Appalachiosaurus*, *Xiongguanlong*, *Timurlengia*, and the Iren Dabasu taxon and the 50% majority rule tree placing the Merchantville taxon alongside *Dryptosaurus* and the clade including *Timurlengia* and the Iren Dabasu taxon as a sister taxon to *Appalachiosaurus* and more derived tyrannosauroids (Fig. S1; *Carr et al., 2017*). Finally, YPM VPPU.021795 was placed in the phylogenetic matrix of *Brusatte et al. (2014a)* to test the relationships of the specimen among Coelurosauria and was found to be the basalmost tyrannosauroid in the strict consensus tree (tree length = 3,250, consistency index = 0.333, retention index = 0.788) (Fig. S2). Thus, there is certainly broader phylogenetic support for the referral of YPM VPPU.021795 to Tyrannosauroidea.

A position outside Tyrannosauridae for YPM VPPU.021795 is also supported by several features outside those coded for in the phylogenetic analysis. For example, the portion of bone on the metatarsals between the medial buttressing surface for metatarsal III and the insertion point for the M. gastocnemius lateralis is flattened, unlike the concave condition found in tyrannosaurids (*Peecook et al., 2014*). To compare YPM VPPU.021795 proportionally with other tyrannosauroids, the robusticity index (RI) used in *Peecook et al. (2014)* was employed. The RI is equal to the ratio of the circumference at the diaphysis to proximodistal length (*Peecook et al., 2014*). The RI for the metatarsal IV included in YPM VPPU.021795 (Table 1) was calculated as 0.2823, appreciably smaller than the indices for tyrannosaurids (0.3200+) (*Peecook et al., 2014*).

## DESCRIPTION

### Metatarsal II

The proximal two thirds of the left metatarsal II (Figs. 1A–1F) is eroded, but the dorsal surface is smooth and partially intact. Measurements of the specimen are in Table 1.

**Table 2** Morphological comparisons between YPM VPPU.021795, *Dryptosaurus aquilunguis*, and *Appalachiosaurus montgomeriensis*.

| Taxon | Metatarsal II, proximal morphology | Metatarsal IV, proximal morphology | Metatarsal IV, distal morphology | Metatarsal IV, size and shaft morphology | Metatarsal II, size and shaft morphology |
|---|---|---|---|---|---|
| *A. montgomeriensis* | Deep, notched proximal articular facet for metatarsal III | Deep, notched proximal articular facet for metatarsal III, semi-ovoid medial outline, dorsoventrally and mediolaterally about the same width (*Carr, Williamson & Schwimmer, 2005*) | Triangular, with a sharp sulcus separating distal hemicondyles figure 19G in *Carr, Williamson & Schwimmer, 2005*) | Elongate, not heavily mediolaterally compressed (figure 19 in *Carr, Williamson & Schwimmer, 2005*) | Elongate, not heavily mediolaterally compressed (figure 19 in *Carr, Williamson & Schwimmer, 2005*) |
| *D. aquilunguis* | N/A | Deep, notched proximal articular facet for metatarsal III, mediolaterally wider than dorsoventrally wide, semi-ovoid medial outline (*Brusatte, Benson & Norell, 2011*) | Triangular, with a slight sulcus separating distal hemicondyles (figure 22F in *Brusatte, Benson & Norell, 2011*) | Elongate, heavily mediolaterally compressed (*Brusatte, Benson & Norell, 2011*) | N/A |
| YPM VPPU.021795 | Shallow proximal articular facet for metatarsal III (somewhat caused by damage and deformation) | Deep, notched proximal articular facet for metatarsal III, semi-ovoid medial outline, dorsoventrally and mediolaterally about the same width | Dorsoventrally widened and rectangular, with a slight sulcus separating distal hemicondyles | Elongate, not heavily mediolaterally compressed | Elongate, heavily mediolaterally compressed (deformation) |

The distal end of the preserved portion of metatarsal II is heavily eroded and apparently deformed from compression, curving laterally toward its distal end. Nevertheless, the insertion scar for the M. gastrocnemius pars lateralis is preserved as a slight, linear fossa on the proximal end of the medial surface of this bone (Fig. 1B) (*Carrano & Hutchinson, 2002*). The proximal articular surface for metatarsal III appears as a ventrolaterally oriented shallow notch. However, the proximal end of this bone is heavily reconstructed (especially at the dorsal end of the proximal surface; Fig. 1C) and likely underwent abrasion and deformation. The proximal articular facet for metatarsal III seems have been somewhat more developed (Fig. 1E), though the preserved bone surface indicates a notch less prominent than in *Appalachiosaurus* and other taxa (Fig. 1E; Table 2; see discussion below). The bulbous morphology of the proximal end may be a taphonomic artifact from transport out to sea and stabilization of this bone during preparation. On the lateral face of metatarsal II, a subtle but dorsoventrally elongate buttressing surface like that observed in the metatarsal II of *Appalachiosaurus* (*Carr, Williamson & Schwimmer, 2005*) begins 50 mm below the lateral facet for metatarsal III and extends downward the shaft, being cut off at the clean break at the distal end of the specimen. The preserved bone of the dorsal half of the proximal end extends medially outward from the shaft 17 mm dorsally, whereas ventrally it only does so by 9 mm. The metatarsal II may have been more robust than the metatarsal IV included in YPM VPPU.021795 when uncrushed, as found in other tyrannosauroids (*Holtz, 2004*).

## Metatarsal IV

The proximal two thirds of the left metatarsal IV (Figs. 2A–2F) is the best preserved portion of the two bones, with little indication of crushing along its shaft. Toward the

bottom of its dorsal surface, the specimen preserves scrapes that are consistent with evidence of shark scavenging (Fig. 2C) found in other specimens of dinosaur from the Atlantic Coastal Plain (*Schein & Poole, 2014*). This evidence of feeding activity by sharks suggests YPM VPPU.021795 floated out to sea before it was deposited. The lateral surface is smooth and has noticeable ventral edge that is almost kinked at a right angle (Fig. 2A). The dorsal surface is smooth and convex, expanding mediolaterally and dorsoventrally outward toward the proximal end of the bone. The ventral surface, which is flattened, mirrors the dorsal surface in its expansion. The medial surface is very flat at the shaft, except for two ridges that extend diagonally from the ventral edge of the medial and lateral surfaces approximately 2 mm above the diaphysis to the dorsal edge of the medial and lateral surfaces at the distal end of the medial face. These form a dorsoventrally elongate, teardrop-shaped buttressing surface for the articulation with metatarsal III (Fig. 2B), a feature found in other arctometatarsalian tyrannosauroids (*Thomson, Irmis & Loewen, 2013*; *Loewen et al., 2013*; *Peecook et al., 2014*; *Brusatte & Carr, 2016*). The elliptical insertion point for the M. tibialis anterior (*Carrano & Hutchinson, 2002*) is faded, but present and undeformed (Fig. 2B). The presence of the undistorted buttressing surface on the medial side with defined borders indicates the bone was not mediolaterally compressed from taphonomy. The proximal expansion of the medial side of metatarsal IV has the deep medial articular facet for metatarsal III found in *Appalachiosaurus*, *Dryptosaurus*, and tyrannosaurids (*Holtz, 2004*; *Carr, Williamson & Schwimmer, 2005*; *Brusatte, Benson & Norell, 2011*). This morphology indicates the autapomorphy of Tyrannosauroidea of a proximally crescentic metatarsal III that is limited to the plantar half of the proximal face of the metatarsus (*Holtz, 2004*). The insertion for the M. gastrocnemius lateralis (*Carrano & Hutchinson, 2002*) is poorly visible due to erosional damage, though it may still be preserved toward the far distal end of the proximal portion of the preserved bone shaft (Fig. 2D). Between the articular surface for the metatarsal III on the medial surface and the insertion point for the M. gastrocnemius lateralis, the shaft of the metatarsal IV is flattened. Previously, the presence of this feature has been used to distinguish between arctometatarsalian non-tyrannosaurid tyrannosauroids and the clade Tyrannosauridae (*Peecook et al., 2014*).

The distal end of this metatarsal IV was also preserved, though it was separated from the main portion of the bone. The measurements of this specimen are included in Table 1. The distal end of metatarsal IV is semi-rectangular rather than triangular in shape and has a subtle sulcus separating the distal hemicondyles (Fig. 2F). The medial face is flattened in distal view, though this may be an artifact of preservation. In lateral and medial views, the distal end is very heavily eroded, especially proximally. The medial collateral ligament pit is semi-ovoid and deeper than the lateral collateral ligament pit. Between this distal end and the proximal portion, the metatarsal is approximately four-fifths complete based on the completeness of the buttressing surface for metatarsal III on the medial face. The estimated length of this metatarsal when complete is 464 mm long proximodistally, approximately the same size as the metatarsal IV of *A. montgomeriensis* (*Carr, Williamson & Schwimmer, 2005*).

## DISCUSSION

### Comparisons with other tyrannosauroids

YPM VPPU.021795 can be confidently assigned to a tyrannosauroid dinosaur based on the presence of an arctometatarsalian metatarsus with a proximally crescentic metatarsal III indicated by the metatarsal IV having a deep, sharply notched proximal articular facet, a metatarsal II with a notch that also indicates close articulation with metatarsal III, the presence of dorsoventrally elongate, teardrop-shaped buttressing surfaces on both bones for the articulation with metatarsal III, and a laterally divergent metatarsal IV distally (*Holtz, 2004*; *Brusatte et al., 2010*; *Loewen et al., 2013*; *Thomson, Irmis & Loewen, 2013*; *Peecook et al., 2014*; *Brusatte & Carr, 2016*).

YPM VPPU.021795 was also compared with other tyrannosauroid metatarsals from Appalachia. In proximal view, the expanded proximal end of the metatarsal II differs from the corresponding surface in the right metatarsal II of *A. montgomeriensis* in that in *A. montgomeriensis*, a far deeper and sharper notch-like articular facet for metatarsal III is present, even accounting for erosional damage and deformation on the Merchantville bone (Fig. 1E; Fig. 19F in *Carr, Williamson & Schwimmer, 2005*). The metatarsal IV of YPM VPPU.021795 lacks the autapomorphic feature found in *D. aquilunguis* of having a shaft that in proximal view has a semi-ovoid cross-section significantly wider mediolaterally than dorsoventrally (*Brusatte, Benson & Norell, 2011*). Taphonomic deformation can be ruled out as the cause for this, as the proximal two-thirds of the metatarsal IV included in YPM VPPU.021795 shows no indication of mediolateral compression from taphonomy. Rather, the proximal shape of this bone in the Merchantville specimen is typical of arctometatarsalian tyrannosauroids (e.g., cf. Fig. 19D in *Carr, Williamson & Schwimmer, 2005*; Fig. 3J in *Thomson, Irmis & Loewen, 2013*; Fig. 1E in *Peecook et al., 2014*). Additionally, the estimated length of the metatarsal IV of YPM VPPU.021795 when complete (Table 1) exceeds the length of the metatarsal IV included in the apparently adult or near-adult holotype specimen of *D. aquilunguis* by 73 mm (*Brusatte, Benson & Norell, 2011*).

In distal view (Fig. 2F), the metatarsal IV is different in morphology from the corresponding element in the holotype of *A montgomeriensis* or *D. aquilunguis* (Fig. 19G in *Carr, Williamson & Schwimmer, 2005*; Fig. 22F in *Brusatte, Benson & Norell, 2011*). Unlike the condition in *A. montgomeriensis* or *D. aquilunguis*, the distal end of metatarsal IV is semi-rectangular rather than triangular in shape and is additionally differentiated from the former in having a more gently arched sulcus separating the distal hemicondyles (Fig. 19G in *Carr, Williamson & Schwimmer, 2005*). As no autapomorphies of *Appalachiosaurus* are found on the metatarsus, YPM VPPU.021795 cannot be confidently assigned to that taxon.

Tyrannosauroid hindlimbs became increasingly robust during ontogeny (e.g., *Currie, 2003*; *Currie, Hurum & Sabath, 2003*; *Carr & Williamson, 2004*; *Carr, Williamson & Schwimmer, 2005*; *Peecook et al., 2014*). However, ontogenetic variation contributing to these differences between YPM VPPU.021795, *D. aquilunguis*, and *A. montgomeriensis* may be ruled out, as the estimated length of the Merchantville metatarsals when complete

greatly exceed the dimensions of the corresponding bones of the holotype individual of *D. aquilunguis*, which is apparently an adult (*Brusatte, Benson & Norell, 2011*), suggesting YPM VPPU.021795 does not represent a growth stage of this taxon and differences between the two specimens are not from ontogeny. In *D. aquilunguis*, the metatarsal IV measures approximately 392 mm long (*Brusatte, Benson & Norell, 2011*; *Peecook et al., 2014*), whereas the estimated length of the Merchantville metatarsal IV is 72 mm longer at 464 mm (Table 1). Though metatarsal length does indeed vary in tyrannosauroids where multiple adult individuals are known (*Peecook et al., 2014*), the disparity in the lengths of the metatarsal IV of *Dryptosaurus* and that of YPM VPPU.021795 (by more than 10% of the length of the latter) along with the morphological differences between them suggest against YPM VPPU.021795 belonging to that taxon. The Merchantville bones do, however, nearly match the dimensions in *A. montgomeriensis* (*Carr, Williamson & Schwimmer, 2005*). This suggests against the hypothesis that the metatarsals represent a different ontogenetic stage of *Appalachiosaurus*, though size has been criticized as an indicator of ontogeny in dinosaurs (*Hone, Farke & Wedel, 2016*).

Individual variation is impossible to rule out, as both *Dryptosaurus* and *Appalachiosaurus* are poorly known outside the holotype specimens of each, making the sample size for each effectively one (*Gallagher, 1993*; *Carr, Williamson & Schwimmer, 2005*; *Brusatte, Benson & Norell, 2011*; *Ebersole & King, 2011*). One metatarsal IV from the Blufftown Formation has been referred to *A. montgomeriensis* (*Ebersole & King, 2011*). Though the specimen figured in *Schwimmer et al. (1993)* closely resembles the metatarsal IV described herein, its estimated length (=440 mm) (*Schwimmer et al., 1993*) is smaller than that for the Merchantville bone by about 30 mm (Table 1). Ontogeny thus cannot be accounted for in this case due to the small sample size and fragmentary nature of the Merchantville specimen and the Blufftown metatarsal. Similarly, comparison of YPM VPPU.021795 with the smaller metatarsals included in AMNH 2550-2553, which belong to either a tyrannosauroid or an ornithomimosaur, is not possible, though it should be noted that YPM VPPU.021795 and these AMNH specimens are separated in time by several million years (*Miller et al., 2004*). In the case of comparison between *D. aquilunguis* and YPM VPPU.021795, the lack of an autapomorphy of the former taxon on the latter specimen is considered beyond intraspecific variation and grounds for separating the two. However, the differences between YPM VPPU.021795 and *Appalachiosaurus*, which include the appreciable difference between their RI, their differently shaped metatarsals IV distally, and the lack of a deepened proximal articular facet for metatarsal III on the metatarsal II of the former taxon, could be from intraspecific variation and are discussed herein.

*Schwimmer et al. (1993)*, *Carr, Williamson & Schwimmer (2005)*, and D. Schwimmer (2017, personal communication) note that the metatarsals of *Appalachiosaurus* and the referred Blufftown specimen match the corresponding bones of *Gorgosaurus libratus* very closely. On the species level, individual tyrannosauroids have proportionally similar metatarsals when such bones are of the same or very similar length (e.g., Fig. 1 in *Currie, 1998*; Fig. 8.11 in *Larson, 2008*; Tables 2, 3 in *Peecook et al., 2014*). For example, several metatarsals of similar length (=±20 mm) assigned to *Albertosaurus*

(AMNH FARB 5232 and AMNH FARB 5233, RIs 0.3513, 0.3568; AMNH FARB 5235 and ROM 807, RIs 0.3972, 0.4106) have RIs that differ by approximately 0.015 or less, an appreciably smaller difference than that of the RIs of *Appalachiosaurus* and the Merchantville taxon (0.0648) or between the latter and *Dryptosaurus* (0.0493) (*Peecook et al., 2014*). Furthermore, the undeformed proximal end of metatarsal IV has a much smaller width dorsoventrally and mediolaterally than *A. montgomeriensis*, 63 to 83 mm and 48 to 82.6 mm, respectively (Table 1; *Carr, Williamson & Schwimmer, 2005*). The metatarsal II of YPM VPPU.021795, though eroded at the proximal end, is also much thinner proximally than the corresponding element in *Appalachiosaurus*, 80 to 100.9 mm dorsoventrally and 55 to 79.5 mm mediolaterally, respectively (Table 1; *Carr, Williamson & Schwimmer, 2005*). Among individuals of *Albertosaurus* and *Tyrannosaurus*, however, the proportions of metatarsals IV of comparable length are similar (*Currie, 1998*; *Larson, 2008*). Additionally, the AMNH *Albertosaurus* metatarsals do not differ in the proximal morphology of their metatarsals II and distal morphology of their metatarsals IV as between YPM VPPU.021795 and *Appalachiosaurus* (e.g., Fig. 1 in *Currie, 1998*; Fig. 19 in *Carr, Williamson & Schwimmer, 2005*). Therefore, the differences between the similarly proximodistally long metatarsals of *A. montgomeriensis* and YPM VPPU.021795 are indicative that the latter is distinct from the former. Differences between YPM VPPU.021795, *Dryptosaurus*, and *Appalachiosaurus* are listed in Table 2.

## Biogeographical considerations

These comparisons (e.g., Table 2) suggest YPM VPPU.021795 represents a distinct taxon of tyrannosauroid on Appalachia. Notably, *Dryptosaurus* and *Appalachiosaurus* have been reported from other Campanian deposits of this landmass (*Baird & Horner, 1979*; *Gallagher, 1993*; *Carr, Williamson & Schwimmer, 2005*; *Brusatte, Benson & Norell, 2011*; *Ebersole & King, 2011*). However, the identification of a new taxon of tyrannosauroid from the Merchantville Formation hints at the possibility that Campanian bones assigned to the two named Appalachian tyrannosauroids could belong to other taxa. For example, teeth and a proximal pedal phalanx collected from the Campanian Ellisdale fossil site were referred to as "cf. *Dryptosaurus*" by *Gallagher (1993)* and "*Dryptosaurus* sp." by *Grandstaff et al. (1992)*, and a tooth from the Campanian of Delaware was also compared to *Dryptosaurus* (*Gallagher, 1993*). More recently, *Denton et al. (2011)* assigned the teeth and limb bone fragment apparently referenced by *Gallagher (1993)* and *Gallagher (1997)* to indeterminate tyrannosauroids, along with several theropod phalanges. The author agrees with this assignment in light of the recognition of a distinct taxon of tyrannosauroid in the Atlantic Coastal Plain. Notably, several partial femora that compare favorably with that of *D. aquilunguis* have been collected from Campanian deposits in North Carolina (*Baird & Horner, 1979*; *Weishampel & Young, 1996*). Thus, it may be concluded that two to three tyrannosauroid taxa were present on Appalachia during the Campanian: that represented by YPM VPPU.021795, *A. montgomeriensis*, and *D. aquilunguis*/cf. *D.* sp.

In the western United States, nine or 10 possible species of Late Cretaceous tyrannosaurs are known, ranging from the Prince Creek Formation of Alaska's north slope to a taxon from the El Gallo Formation of Mexico (*Lambe, 1917*; *Holtz, 2004*; *Weishampel et al., 2004*; *Loewen et al., 2013*; *Thomson, Irmis & Loewen, 2013*; *Fiorillo & Tykoski, 2014*; *Peecook et al., 2014*; *Carr et al., 2017*). This range is obviously larger latitudinally, as the main Late Cretaceous outcrop from Appalachia ranges only from New Jersey to the southern states of Alabama, Mississippi, and Georgia and westward to southeastern Missouri (*Schwimmer, 1997*, *2002*; *Fix & Darrough, 2004*). Thus, the number of tyrannosauroid taxa from Appalachia is not directly comparable to that from Laramidia, especially when the significant taphonomic biases against the preservation of dinosaurs from Appalachia are considered (*Schwimmer, 1997*, *2002*). There are presently six named tyrannosauroid species from the Campanian of the western United States (excluding Alaska) (*Bistahieversor sealeyi*, *Daspletosaurus torosus*, *Daspletosaurus horneri*, *G. libratus*, *Lythronax argestes*, *Teratophoneus curriei*) (*Lambe, 1917*; *Russell, 1970*; *Weishampel et al., 2004*; *Carr & Williamson, 2010*; *Carr et al., 2011*; *Loewen et al., 2013*; *Carr et al., 2017*), though *Thomson, Irmis & Loewen (2013)* and *Peecook et al. (2014)* described hindlimb elements of possibly distinct Campanian Laramidian tyrannosaurid taxa. Though the diversity of the clade on these two landmasses is hardly comparable at this date, the author would not be surprised if future review of Campanian Appalachian tyrannosauroid material yields higher diversity estimates for the clade on the eastern landmass.

Finally, the results of the phylogenetic analysis of Tyrannosauroidea including YPM VPPU.021795 and the morphology of the bones suggest the Merchantville taxon was an "intermediate" tyrannosauroid, supporting the hypothesis that Appalachian dinosaur faunas contained relict forms (*Schwimmer, 1997*). *Brusatte, Benson & Norell (2011)* discussed the possibility of a distinct eastern clade of Late Cretaceous tyrannosauroids. Although the assignment of YPM VPPU.021795 to another one of what *Brusatte, Benson & Norell (2011)* termed "intermediate" tyrannosauroid taxa provides further support that such forms were somewhat diverse on Appalachia, the phylogenetic analysis herein did not place *Dryptosaurus*, *Appalachiosaurus*, and the Merchantville taxon in a distinct clade (Figs. S1 and S2). Thus, there is currently little phylogenetic support for a distinct group of tyrannosauroids on this Cretaceous landmass.

## CONCLUSIONS

A tyrannosauroid metatarsus from the Campanian Merchantville Formation of the Atlantic Coastal Plain can be reasonably distinguished from the two named Appalachian tyrannosauroid taxa (*A. montgomeriensis* and *D. aquilunguis*), thus representing a potentially distinct taxon of tyrannosauroid in the Campanian of eastern North America. Based on the incomplete nature of YPM VPPU.021795, the author does not think it wise to name a new taxon based on this specimen. However, the specimen increases the known diversity of Appalachian tyrannosauroids and also supports the previously stated hypothesis that Appalachian dinosaur faunas included relict forms that survived on the landmass throughout the Late Cretaceous.

## INSTITUTIONAL ABBREVIATIONS

**AMNH FARB**    Fossil amphibian, reptile and bird collections of the American Museum of Natural History, New York, NY, United States.

**ROM**    Royal Ontario Museum collections, Toronto, ON, Canada.

**YPM VPPU**    Princeton University collection in the Division of Vertebrate Paleontology, Yale Peabody Museum, New Haven, CT, United States.

## ACKNOWLEDGEMENTS

I thank Daniel Brinkman for allowing access to view and describe YPM VPPU.021795 in the Yale Peabody Museum collections and Carl Mehling for allowing access to view and describe AMNH FARB 2550-2553. I thank David Schwimmer for providing information on the metatarsus of *Appalachiosaurus* and Mark Loewen for providing me with a copy of the matrix of *Loewen et al. (2013)* for use with TNT v.1.5 for an earlier draft of this manuscript. I thank the academic editor Andrew Farke and reviewers Stephen Brusatte, Alexander Averianov, and Thomas Carr for their comments and suggestions, which all were extremely beneficial in improving the quality of this paper. Finally, I thank the commentators on the preprint of this article for discussions on the specimens and acknowledge the Willi Hennig Society for sponsoring the TNT phylogenetic software used herein (*Goloboff & Catalano, 2016*). When this manuscript was in peer review, the author received notice that another research group was also working on the specimens and had submitted their manuscript for peer review. The author looks forward to reading their manuscript and comparing the results of both studies.

### Funding
The author received no funding for this work.

### Competing Interests
The author declares that they have no competing interests.

### Author Contributions
- Chase D. Brownstein conceived and designed the experiments, performed the experiments, analyzed the data, contributed reagents/materials/analysis tools, wrote the paper, prepared figures and/or tables, reviewed drafts of the paper.

### Data Availability
YPM VPPU.021795 is stored in the collections of the Yale Peabody Museum in New Haven, CT.

### Supplemental Information
Supplemental information for this article can be found online at http://dx.doi.org/10.7717/peerj.4123#supplemental-information.

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
