# Peer review of "A tyrannosauroid metatarsus from the Merchantville Formation of Delaware increases the diversity of non-tyrannosaurid tyrannosauroids on Appalachia"

_PeerJ, doi:10.7717/peerj.4123_

## Round 0.1 · original submission · Major Revisions

· Academic Editor

Major Revisions

- The reviewers had particular concerns about the identification of the elements discussed here--e.g., could they be ornithomimid or something else? Are ontogenetic effects at play? These alternative (or supplementary) interpretations must be addressed in revision. Along these same lines, the reviewers suggest a phylogenetic analysis might help solidify the identification of the material. Although it is quite incomplete, such an analysis would help to better constrain phylogenetic placement.
- The reviewers provide a number of other thoughtful suggestions that should be considered in revision. Given the scope of revision required, an addition round of review will likely be required.

·

Basic reporting

see below

Experimental design

see below

Validity of the findings

see below

Comments for the author

This paper describes two new dinosaur specimens from the Late Cretaceous of New Jersey. Dinosaur material remains very rare in the eastern United States, so any new specimens are noteworthy and deserve publication, particularly as many of them are latest Cretaceous in age, so have relevance for understanding dinosaur diversity and biogeography during the run-up to the mass extinction. I have worked on several of the east coast theropod specimens, particularly the tyrannosauroids, and so I am happy to see new material discovered and described. There is no doubt that these specimens are worthy of publication in the scientific literature.

However, I cannot yet recommend publication of this paper. The presentation and content require major revisions. Here are my main issues. I hope you can address them, and at that point, the paper will be ready for publication in PeerJ:

1) More information is needed on how the specimens were found. Were they found associated? How close together were they, and in what orientation? What else was found with them? How do we know they belong to the same individual? Particularly as the metatarsal IV has marks indicative of shark feeding traces, and the specimens are found in a marine unit, meaning they must have been transported far from the site of death.

2) You need to make very clear which features support a tyrannosauroid affinity, rather than, say, an ornithomimosaur affinity. The best way to do this is to include them in a broad phylogenetic analysis of coelurosaurian theropods, like the Theropod Working Group matrix (most recent version: Brusatte et al. 2014, Current Biology).

3) Is the metatarsal II actually a metatarsal II? In proximal view there are notches on both lateral and medial surfaces, which is not the case of metatarsal II in tyrannosauroids (there is only a notch on the lateral surface, where metatarsal III articulates). What would be articulating with the medial notch if this is a metatarsal II? There is nothing there in tyrannosauroids. Also, the proximal end is much more bulbous compared to the shaft than in other tyrannosauroids. Are you sure this is not a metatarsal III of a non-arctometatarsalian animal?

4) On first glance, the specimens (especially the metatarsal II) do not strike me as obviously tyrannosauroid. They seem to be a little smaller and more gracile than the metatarsals of Appalachiosaurus/Dryptosaurus-like tyrannosauroids, not to mention the larger and more robust Tyrannosaurus-like species. One possibility is that they belong to an ornithomimosaur (see above). The other is that they belong to a juvenile. Therefore, comparisons with juvenile tyrannosauroids are needed. If the specimens belong to juveniles, ontogenetic variation may explain some of the differences with Dryptosaurus and Appalachiosaurus (both of which are known from larger, presumably more mature, holotypes).

5) The specimens seem somewhat deformed. Could this have caused some of the subtle shape differences compared to Appalachiosaurus and Dryptosaurus? I agree that metatarsal IV does not have the autapomorphically ‘flattened’ condition of Dryptosaurus (in which the proximal articular surface is much wider mediolaterally than long anteroposteriorly). However, deformation must be ruled out as the reason for this difference. The same is true of other observed differences.

6) You simply assert that the few differences between your new specimens and Appalachiosaurus and Dryptosaurus cannot be the result of individual variation. You have to back this up with evidence. What type of individual variation is present in the metatarsals of well-sampled tyrannosauroids, such as Tyrannosaurus, Tarbosaurus, Albertosaurus, or Gorgosaurus? And more importantly: what type of ontogenetic variation is present in these taxa?

7) The treatment of the ‘unnamed specimen from the Maastrichtian of New Jersey’ is problematic. This specimen is alluded to throughout the text without mention of its specimen number, until that number is finally given in the discussion. This is AMNH 2550-2553. It is not a tyrannosauroid, but rather, an ornithomimosaur. I suggest that you simply ignore this specimen for the time being and leave it out of your paper. It isn’t very relevant to the material you are describing.

On that note, you say that you are currently writing up a description of this specimen. You may not be aware, but my colleagues and I (Brusatte, Benson, Choiniere, Norell) began studying this material in 2012 and wrote an SVP abstract about it, which was peer-reviewed and presented at the 2012 Raleigh meeting. I respectfully ask you to respect the priority we have established in describing and naming this material. We will be writing a paper on it; we already have a draft manuscript.

8) I agree that the distal end of metatarsal IV is different in shape to that of Appalachiosaurus, as you describe. However, you also need to describe how it is different in shape to that of Dryptosaurus. You simply assert this without delimiting the specific anatomical features that differ.

9) The figures need to be labelled—point out the most important anatomical features.

10) What would really beef up this paper, and drive home the differences with Appalachiosaurus and Dryptosaurus, are comparative figures and tables. Show photographs of the new specimens alongside the same bones of the Appalachiosaurus and Dryptosaurus holotypes, and also present a table of measurements for the specimens.

11) You should include the new specimens in a phylogenetic analysis. There are very detailed, comprehensive phylogenetic studies of tyrannosauroids in the literature, which use vast datasets of anatomical characters including many features of the metatarsus. The most comprehensive is the recent Brusatte & Carr analysis (2016, Scientific Reports), which is a follow-up of an earlier study (Brusatte et al. 2010). You could simply score the new specimens for this dataset, run the analysis, and see where it is placed on the cladogram.

In that vein, I was very surprised to see that these two studies weren’t even cited to support the following statement: ‘The tyrannosauroid taxa of this landmass, Dryptosaurus and Appalachiosaurus, have repeatedly been found outside Tyrannosauridae in phylogenetic analyses’. They need to be read, considered, and cited.

12) You consider Dryptosaurus as present in both the Campanian and Maastrichtian, and even as present in the same formation as the new specimens. Be careful here. Yes, several specimens from across eastern North America have been referred to Dryptosaurus, but this has not been based on autapomorphies, but rather on general similarity. The holotype of Dryptosaurus is Maastrichtian in age. Do any of these other specimens share unique features with the holotype? If so, then you can confidently refer them to Dryptosaurus. If not, then you should probably consider them Tyrannosauroidea indet. This is important, because the identification of these specimens factors into your biogeographic discussion and conclusions. In fact, your biogeographic discussion hinges, in a sense, on whether or not these specimens are truly Dryptosaurus or belong to another taxon.

13) I think it is way, way too premature to argue that Appalachian tyrannosauroids had a similar diversity to Laramidian tyrannosauroids. I am not convinced that what you have here is a ‘new morphotype’ rather than an individual variant of Appalachiosaurus, or possibly Dryptosaurus. But even if it is a new taxon, that means there are three known tyrannosauroids from the entirety of the Campanian and Maastrichtian of Appalachia. That is far, far fewer than the known number from the entire Campanian and Maastrichtian of Laramidia. Of course, there are huge sampling biases—much less is known about Appalachian dinosaurs, because much less outcrop is available for study. But, at this point in time, there is simply no positive data to suggest that Appalachia ‘probably enjoyed similar (tyrannosauroid) diversity’ to Laramidia, as you conclude in the discussion. It is an hypothesis that can be tested by future discoveries, but we certainly cannot conclude that it is correct at this point!

14) Finally, there are many misspellings and grammatical issues throughout—e.g., autopomorphy instead of autapomorphy. The references are presented in a random, non-alphabetical order. There are also some basic anatomical errors in this paper. Metatarsal II is said to have an articular surface for metatarsal III in medial view; however, it would be the lateral surface of metatarsal II that articulates with metatarsal III. The manuscript needs a careful edit to reach the standard needed for the peer-reviewed literature.

Steve Brusatte, Univ of Edinburgh, July 23, 2017

·

Basic reporting

OK

Experimental design

OK

Validity of the findings

Identification of the bones as belonging to Tyrannosauroidea is not well supported. Alternative identifications are not considered.

Comments for the author

The paper describes partial theropod metatarsus (proximal fragments of metatarsals II and IV) from the Campanian Merchantville Formation of New Jersey, USA. Because of extreme rarity of theropod dinosaurs from Appalachia, this funding represents a considerable interest. The author referred the specimen to the Tyrannosauroidea based on metatarsal III (reconstructed) of crescentic shape and restricted to the plantar surface of proximal end of metatarsus. These characters are considered by author following Holtz (2004) as autapomorphies for the Tyrannosauroidea. However, these characters are found also in the Ornithomimidae. The author did not discuss the alternative identification of the described specimen as an ornithomimid and did not rule out this possibility. However, described bones show considerable similarity with some ornithomimids (see, for example, a well described metatarsus of Qiupalong henanensis; Xu et al., 2011). By gracile nature I think that the described metatarsals from the Merchantville Formation are more referable to an ornithomimid rather than to the tyrannosauroids, which had massive and robust metatarsals. The author violates the International code of Zoological Nomenclature (Article 36). Some other comments are introduced in the attached pdf.

Reviewer 3 ·

Basic reporting

The prose could really be cleaned up, especially with regard to passive voice and redundancy.

Literature references are incomplete and insufficiently used.

Professional article structure is present.

More measurements could be included.

Self-contained and the results are relevant to hypotheses, although overstated and incomplete in places.

See general comments and confidential note for details.

Experimental design

original primary research is present, although arguably incomplete.

Research question is defined, but more thorough comparisons could be made.

The knowledge gap is incompletely filled.

Ethical, but not rigorous.

Methods are described and replicable.

See general comments and confidential note for details.

Validity of the findings

Impact and novelty could be improved with additional study of specimens and literature.

Data are incomplete.

Conclusions are overstated in places.

See general comments and confidential note for details.

Comments for the author

Dear Editor,

I submit to you my review of “A tyrannosauroid metatarsus from the Merchantville Formation of New Jersey increases the diversity of non-tyrannosaurid tyrannosauroids on Appalachia” by Brownstein (ms #19258). The article describes the partial metatarsus of a basal tyrannosauroid in comparison with other taxa and isolated bones, all from the late Cretaceous of Appalachia.

My opinions on the ms are mixed, as I am happy to see an additional specimen added to the roster of tyrannosauroids from the Appalachia, but the description strikes me as incomplete. In particular, the comparisons of the new metatarsus with other Appalachian taxa appear to be fairly superficial, perhaps owing to the fact that only two museum collections were visited. To really nail down the comparisons, all of the relevant specimens must be examined first hand and, ideally, side by side. For example, it appears that the holotype of Appalachiosaurus was not view first-hand, and comparisons were only made from the published literature.

I am not convinced that the author could really determine that “the spectrum of individual variation” of Appalachian tyrannosaurids was truly identified, since the relevant taxa are represented by only one specimen. Also, the author did not take advantage of the recent phylogenetic work on tyrannosauroids (e.g., Brusatte and Carr, 2016; Carr et al., 2017), where character states of the metatarsus are listed that could have helped flesh out the comparisons and characterizations of the new material and the other species.

The figures could be more fully labeled so that they correspond to the observations made in the text. The captions are incomplete for metatarsal IV.

I think a table that summarizes all of the major similarities and differences between the new metatarsus and other Appalachian taxa would greatly benefit the manuscript.

The prose could really be cleaned up, especially with regard to passive voice and redundancy.

The biogeographic conclusions are somewhat overstated and can be simplified to just the facts.

In conclusion, I think the author could satisfactorily make the necessary revisions and so I recommend “major revision”.

The author may know my identity: Thomas D. Carr.

My specific comments are below.

Sincerely,

Thomas D. Carr, PhD
Associate Professor of Biology
Carthage College

Specific comments:

Lines 7-10: overuse of the word “obscure” and its variants.
Lines 9-10: “do seem to be”; passive voice – fix.

Lines 10-11: “significant”; this word has statistical meaning and should not be used informally here.

Lines 18-20: the spectrum of variation cannot be known for taxa that are represented by only one specimen – please fix.

Lines 19-20: replace “as well as by factoring in” with “and”.

Lines 19-20: replace “The new specimen thus has significance for representing” with “Therefore, the new specimen represents”.

Lines 21-25: convoluted – just state the result.

Last sentence of abstract: change to “Appalachian non-tyrannosaurid tyrannosauroids had a diversity that was comparable to tyrannosaurids in Laramidia during the Campanian”.

Line 31: in what way is Appalachia “very poorly known”? Clarify.

Line 31: replace “comparison” with “contrast”.

Line 40: delete “repeatedly”.

Lines 42-44: how is all of that unfortunate? Clarify.

Line 52: delete “described herein thus”.

Line 53: insert “a new,” between “possibly” and “unnamed”.

Line 53: delete “dinosaur”.

Line 54: replace “. Thus, the specimen adds to the diversity of the clade Tyrannosauroids” with “, adding to the diversity of the the clade”

Line 55: replace the comma with a period.

Line 55: replace ”suggesting” with “The specimen is evidence that”

Lines 56-57: replace “the aforementioned” with “that”.

Line 61: replace “the described” with “this”.

Line 67: in my experience, the full AMNH specimen number abbreviation is “AMNH FARB”.

Line 77: delete “described herein”.

Line 77: replace “retrieved” with “collected”.

Lines 77-81: were the two bones found together? State explicitly.

Line 79: replace the ampersand with “and”.

Lines 81-85: any radiometric dates? If so, please include here.

Lines 98-104: what are the sizes of the specimens?

Line 107: replace “of metatarsal II with” with “for”, and label the feature in the figure.

Lines 110: explain what is meant by “a more prominent articular facet”; this is vague – be exact.

Line 111: delete “exists”.

Line 111: insert “Therefore,” before “The”.

Line 112: delete “thus”; “not as distinct” is vague – be exact; delete “in medial view”; “and,….metatarsal II.” is convoluted – fix.

Line 115: replace “slight” with “shallow” and label it in the figure.

Line 118: delete “as measured along the proximal surface”.

Line 119: replace “better” with “best”.

Line 120: replace “metatarsls” woth “bones”; insert “that are” after “scrapes”.

Line 124: replace “confirms” with “is evidence”; replace “being” with “it was”; delete “and preserved”.

Line 125: “a sharp ventral edge” is vague – clarify and label in the figure; replace “rounded” with “convex”.

Line 126: is the ventral surface flat? Clarify.

Line 128: “just above the diaphysis” is vague – clarify.

Line 133: replace “suggests the presence of” with “is evidence for”.

Line 134: delete “having”.

Line 135: “plantar” – be consistent with terminology throughout the ms.

Line 136: what is the specimen number of the unnamed specimen? Mention it here.

Line 136: replace “also…specimen” with “less pronounced than in YPM…obs.); delete “distinguishing the two specimens”.

Line 138: delete “unnamed New Jersey specimen’s”.

Line 139: insert “of New Jersey” after “IV”; “a much thinner shaft” is vague, provide measurements; delete “(pers. obs.)”.

Line 140: deete “notably”.

Line 143: replace “further be” with “also be”, delete “in…latter”.

Lines 149-152: Delete “In…2011).”

Line 154: replace “an even less noticeable” with “more aubtle”.

Line 161: give the estimated total length here.

Line 167: “temporslly differentiated” is vague – be exact.

Line 171: give the measurements instead of giving “pers. obs.”

Lines 172-173: list all of the differences in a new paragraph here or provide a table of comparisons and differences.

Line 173: delete “morphological” from here and elsewhere.

Line 175: replace “element” with “bone”.

Line 178: “less robust” is vague – be exact.

Line 182: delete “the assignment of”, insert “is assigned to” after “21795”; delete “is considered best”.

Line 184: list the comparisons!

Line 184: delete “and this indeterminate taxon,”; delete “dinosaur”.

Line 186-187: fix the sentence that starts with “The identification”.

Line 188: replace “from” with “in”; replace “corresponding to this landmass” with “of Appalachia”.

Line 196: replace “a possibly” with “an”, replace “animal” with “taxon”.

Line 201” replace “tyrannosaur” with “tyrannosauroid”.

Lines 205-206: delete “especially…account”.

Line 207: replace “enjoyed” with “had a”.

Lines 210-212: convoluted – fix.

Line 217: what are the differences? List them here.

Line 218: replace “currently…author” with “in progress”.

Lines 218-219: what are they? Be exact.

Lines 221-237: doesn’t all of this boil down to a sampling issue? Delete.

Line 246: delete “from the lower 48”.

Line 247: replace “tyrannosaur” with “tyrannosauroid”.

Lines 248-250: Delete.

Line 254-255: replace “. Additionally thank” with “, and”.

Lines 255-256: delete “in…History”.

Figure 2 caption: the caption is incomplete – what is the arrow pointing to? The distal condylar region is not included.

---

## Round 0.2 · Major Revisions

· Academic Editor

Major Revisions

I have looked over the revision, and find that it addresses many of the concerns raised by the reviewers. Prior to doing so, though, several aspects of the phylogenetic analysis should be considered:

1) The full phylogenetic dataset should be included as a TNT file to more easily allow reviewers and readers to test your results.

2) A figure of the consensus phylogeny should be included, perhaps as supplemental information.

3) Finally, I note that the Brusatte and Carr matrix is not sufficient to thoroughly test the hypothesis of tyrannosauroid affinities. Their matrix focused on tyrannosauroids, so it is cannot adequately compare competing interpretations. I would suggest, as suggested by one of the original reviewers, running the specimen also in a broader matrix (e.g., the Theropod Working Group matrix, or a similar matrix) that includes ornithomimids, oviraptorosaurs, etc. I would still include the tyrannosauroid matrix; just add one more analysis of a matrix with broader taxonomic sampling.

Once these concerns have been addressed, I will be able to send the manuscript out for a second round of evaluation by the reviewers.

---

## Round 0.3 · Minor Revisions

· Academic Editor

Minor Revisions

Thank you for your close attention to the previous round of suggestions from the reviewers. I sent the manuscript out for one additional round of review; we have a handful of comments remaining to address. These are outlined elsewhere in this email and at the PeerJ website.

- re: the comparisons with Appalachiosaurus, I agree with the reviewer that a bit more discussion of how any deformation/incompleteness may affect your interpretations would be desirable. Firsthand comparison of the fossils would also be useful, but I do not mandate this (particularly given the costs and time involved, and considering the comprehensive figures already published in the original Appalachiosaurus paper).

·

Basic reporting

see below

Experimental design

see below

Validity of the findings

see below

Comments for the author

I thank the author for taking into account all of my suggestions and carefully revising the manuscript. I may have a few quibbles about some of the anatomical details, but at this point, I think the author has been very rigorous and deserves to have this manuscript published so others can assess the specimen.

One thing I would ask the author to do: explain in more detail the phylogenetic analysis and how it was conducted. It is said that a ‘strict consensus analysis’ was run in TNT. This doesn’t make sense. I presume the author means that the analysis was used to generate a strict consensus phylogeny of the most parsimonious trees, after the analysis was conducted. So the author needs to be clear on what protocols were used in the analysis: New Technology search (and the specs)?, TBR branch swapping (and the specs?) etc. The author should try to follow the exact protocol that we outlined in the Brusatte & Carr (2016) paper, to make it maximally comparable with our analysis.

Reviewer 3 ·

Basic reporting

In many places the prose is convoluted and compromised by redundancy and passive voice.

The literature references are sufficient.

Sufficient background and context is given.

A professional article structure is followed, although subheadings for comparisons with other taxa might be helpful.

The article is self-contained, although I think that the issues of individual and ontogenetic variation can be carried out more rigorously and concisely (see comments to author).

The author may know my identity: Thomas D. Carr, PhD

Experimental design

The article is within the aims and scope of the journal insofar as evidence for a new taxon of tyrannosauroid is presented, although based on a poorly preserved specimen.

The research question is straightforward, where the author presents evidence for a new taxon of Campanian tyrannosauroid; given the limited nature of the fossil, it fills a very small gap in our knowledge.

However, I am concerned that it hasn't been compared side-by-side with Appalachiosaurus, especially given that some differences between them are the result of damage to the Merchantville specimen (see ms Table 2).

I don't think the comparisons made are rigorous (see comments to author), but these can be improved.

The methods are adequately described and replicable, although I would run the Merchantville specimen in the latest TWiG matrix, not the Loewen et al. (2013) matrix.

The author may know my identity: Thomas D. Carr, PhD

Validity of the findings

Given the nature of the Merchantville fossil, and the limited scope of the paper, the impact is not great beyond the addition of a new tyrannosauroid from Appalachia. Not much can be said beyond that.

The novelty is limited to the possibility of a sympatric tyrannosauroid with Appalachiosaurus, but given the wide range of the age of the Merchantville Formation, they might not really have lived at the same time.

The Merchantville specimen does not reveal anything new about tyrannosauroid evolution, aside from increasing its diversity by one taxon.

The data are robust, but the RI for the Merchanville taxon is compromised by deformation (table 2), which reduces my confidence in its difference from Appalachiosaurus.

The conclusions are fine, but the closing statements regarding incomplete specimens isn't needed since the limited amount of information they contain is a given.

Speculation is thankfully avoided.

The author may know my identity: Thomas D. Carr, PhD

Comments for the author

October 9, 2017

Dear Dr. Farke,

I submit to you my review of the second draft of “A tyrannosauroid metatarsus from the Merchantville Formation of Delaware increases the diversity of non-tyrannosaurid tyrannosauroids on Appalachia” #19258, by C. D. Brownstein. The author has taken care of many of the issues that I raised in the first draft, but some nagging problems remain: the prose is compromised by passive voice and excess verbiage, the introduction needs to be almost entirely redrafted, first-hand comparisons have not been made with Appalachiosaurus, and the discussions of individual and ontogenetic variation should be more rigorous.

The bottom line is that the sample sizes for the three taxa of concern are, effectively, one each; therefore, the only resort to clarifying individual and ontogenetic variation is for the author to examine a growth series of metatarsals of a known tyrannosaurid. Such a growth series is available at the AMNH, based on the Albertosaurus bonebed material. Using that, the author can parse out the features that do change with size (as a proxy for maturity, size is not the best option - I know - but the gross patterns should be clear) and those that don’t, and then compare the corresponding features with the other Appalachian fossils to identify which ones are taxonomically informative and which aren’t. If this was my project, that’s what I’d do.

If this procedure is followed, then the author would arrive at defensible, empirically based statements such as:

“The RI does not change in the growth series of A. sarcophagus, where it is 0.33 in all specimens. The RI of the Merchantville specimen is less than this and that of Appalachiosaurus; therefore, it is a different taxon from them. The RI of Dryptosaurus is 0.15; therefore, the Merchantville specimen is not referable to Dryptosaurus. Given these differences, the Merchantville metatarsus represents a new morphotype, and hence a new taxon.”

Or,

“In end view, the distal end of metatarsal IV is triangular and rectangular in the growth series of A. sarcophagus, without any pattern. Therefore, the triangular shape of the joint surface in the Merchantville specimen is taxonomically and ontogenetically uninformative, and so it is not a reliable point of comparison with Dryptosaurus and Appalachiosaurus.” And so on.

Unfortunately, I still think substantial revisions must be made. Please see my specific comments below for more areas of improvement.

Sincerely,

Thomas D. Carr, PhD
Associate Professor of Biology
Carthage College
Kenosha, WI 53140


Specific comments

Abstract
Line 11: replace “this latter landmass” with “Appalachia”.
Line 12: replace “of….apparently” with “were”.
Lines 12-13: replace “Laramidian ones” with “Laramidia”.
Lines 13-15: these sentences can be shortened and combined into one.
Lines 17-19: The justification for giving the metatarsus such attention must be stated here: is it well-preserved enough to bring new insights into Appalachian tyrannosauroid diversity and evolution?
Line 19: replace “may” with “can”.
Lines 21-23: please list the defining features here; otherwise this sentence is just indicative.
Line 24: replace “morphotype” with “taxon”.
Lines 25-26: adding a new taxon is not a test of the hypothesis that Appalachia was a refuge of relict taxa – it is just evidence of a new taxon. The evidence for a relict taxon is its phylogenetic position; so, if it falls out in a cladistic analysis outside of Bistahieversor, then you have evidence for a basal, relict tyrannosauroids. Otherwise, you don’t. I.e., please state the results of your phylogenetic analysis in the abstract.

Introduction
Lines 29-32: this first sentence is unnecessary – just get to the point, which is in the second sentence.
Line 46: keep in mind that we didn’t run a Bayesian analysis in Carr et al. (2017); the results published there were from a parsimony analysis.
Lines 46-49: redraft this sentence as “The fossil record of tyrannosauroids from Appalachia is limited to the holotypes of Dryptosaurus and Appalachiosaurus, undescribed partial specimens, and isolated bones and teeth.”
Lines: 50-88: This section is written like the discussion and conclusions, not a proper introduction that justifies the project, defines the research question and its importance and novelty, describes the methods and the main objectives of the project. Redraft.

Methods
Lines 92-95: As with the earlier draft, the author has not seen the holotype of Appalachiosaurus first hand.

Material
Lines 136-157: please restructure this by first stating the arctometatarsalian features that do not help to identify the taxon, followed by the unambiguous tyrannosauroid characters.
Lines 158-180: it isn’t at all clear if these features are seen in the specimen or not; for each feature, describe what is seen in the purported tyrannosauroid so that the reader can clearly see what features you are eliminating and why.

Lines 181-184: this is the biggest blunder of the article, given that Brusatte and Carr (2016) combined the data sets of Loewen et al. (2013) and Brusatte et al. (2010) to resolve the huge differences in results obtained from those data sets.

Therefore, scoring the specimen into the Loewen et al. matrix makes no sense on theoretical or practical grounds – it is just irrelevant. In fact, Carr et al. (2017) updated the Brusatte and Carr (2016) data matrix with additional characters (and taxa), so that should be the data matrix used to assess the affinities of the metatarsus, if it is a tyrannosauroid. Please redo the phylogenetic analysis using the most complete and up-to-date data set.

I also recommend scoring the specimen in the latest version of the TWiG (Theropod Working Group) data matrix to assess the identity of the metatarsus among Theropoda as a whole.

Line 181: delete the underscore in the specimen number.
Line 184: insert a space between the parentheses.

Description
Line 187: delete “noticeably”.
Line 189: replace “may be found” with “are”.
Line 190: is it deformed by the erosion or by something else? Clarify.
Lines 187-275: this part of the manuscript is rife with passive voice; please fix, which will help to shorten the text somewhat.
Lines 207-234: these sections are a return to assessing the taxonomic identity of the specimen, which is confusing. Please move the relevant parts up to where the identification as a tyrannosauroid is justified and referral to Ornithomimosauria is rejected.
Line 235: perhaps each metatarsal should get their own subheadings in the description to aid the reader.
Line 250: a comma usually precedes the words “but” and “which”.
Line 274: insert “long” after “mm”.

Discussion
Line 295: it is said here that the specimen was coded into the matrix of Brusatte and Carr (2016), whereas earlier it was stated that it was scored for the Loewen et al. (2013) data matrix – which is it? Please fix.
Lines 302-304: this description of the topology is ambiguous – how can the metatarsus and Bistahieversor both be the sister taxon of Tyrannosauridae? Please clarify and include a figure of the results.
Lines 304-306: analysis in the Loewen et al. matrix is unnecessary, as explained above.
Line 307: state the characters that support the position of the metatarsus on the topology before mentioning features that were not included in the data matrix.
Lines 310-314; please state how the RI was obtained, state the result, and give some indication of how it is different from tyrannosaurids.
Lines 315-317: this introductory sentence is convoluted; please rephrase concisely.
Line 330: the phrase “Regardless of the ontogeny” is confusing, as what is meant by that is not explained. Please simplify this sentence to refocus it on the comparison with Dryptosaurus.
Lines 333-335: make sure that this sentence is in reference to the Brusatte and Carr (2016) data matrix, because as stated here you could be talking about that analysis or the Loewen et al. results. Please fix.
Lines 336-341: both of these sentences say the exact same thing – get rid of one of them!
Line 349: the sentence “Description…author.” adds nothing; please delete.
Lines 347-360: I’d consider discarding this section, since you do not unambiguously assess the taxonomic identity of the AMNH specimens.
Lines 365-370: I’d rephrase this to say that the Merchantville metatarsus is approximately the same size as the Dryptosaurus and Appalachiosaurus fossils, and the differences between them are almost certainly not ontogenetic since the bones in the same taxon of the same ontogenetic stage would converge upon the same morphology.
Lines 361-412: this is an unacceptably long paragraph; I recommend breaks at lines 371, 378, 384, 393, 403.
Lines 371-373: the sample size for these taxa is effectively one each – please be exact to convey the point that individual variation is simply impossible to document for them.
Lines 376-378: size is a poor proxy for relative maturity, so the real limit on assessing ontogeny is the same for that of individual variation – you’d don’t have a adequate sample to work with. Fix.
Lines 382-384: You should compare the Merchantville specimen side-by-side with Appalachiosaurus in person. Briefly remind the reader here of what the difference are between the two.
Lines 384-390: I don’t find the argument of geographic separation to be convincing, especially when both taxa are on the one and the same landmass.
Lines 387-390: this sentence is too convoluted; be concise.
Lines 384-413: I think that all of this can be boiled down to one paragraph that summarizes the salient differences between the Merchantville specimen and Appalachiosaurus, excluding the biogeographic argument. What would potentially seal it is comparisons of the RI of metatarsals of the same species of tyrannosaurid that are the same length.
Line 403: please define what the RI is, and what measurements are used.
Line 407: please include the RIs here for T. rex and A. sarcophagus. Please include the RI for Dryptosaurus.
Line 408: delete “the author regards”.
Lines 408-411: the only relevant line of evidence to separate the two is the morphological differences. Revise.

Biogeography
Line 417: replace “morphotype” with “taxon”.
Lines 421-423: this sentence is redundant; delete and modify the following sentence accordingly.
Lines 429-430: delete this sentence; it does not contribute any meaningful information, especially since there are no overlapping bones with the Merchantville metatarsus.
Line 433: you have provided evidence for increased diversity; please delete.
Line 438: this could be changed to “cf. D. sp.”.
Line 451: insert a space between the parentheses.
Lines 457-460: restate this as a specific, defensible hypothesis or delete.
Lines 461-465: convoluted and excessively long sentence – please trim down and clarify.
Lines 469-471: there isn’t a clade – the Appalachian taxa form a grade, so there’s nothing to name.

Conclusions
Line 475: I think it is acceptable to say the morphotype is a new taxon, which you have wisely not named.
Line 477, and above biogeography discussion: I recommend that you avoid discussing the affinities of the other Appalachian material unless they are metatarsals. I think that teeth and postcranial bones that are not metatarsals are beyond the reach of what you actually have described. It is sufficient to mention those fossils, but it is important to point out that their identity cannot be assessed based on the Merchantville metatarsals, since none of them are associated with a metatarsus.
Lines 481: I’d restate this in terms of the challenge of identifying isolated bones that that autapomorphies.

Acknowledgements
Lines 487-493: make sure that you thank your reviewers by name.

Figures
The font size of the labels are different; please standardize.
I recommend adding a comparative figure of the Merchantville specimen side-by-side with the corresponding bones from the holotypes of Dryptosaurus and Appalachiosaurus.

Table 1.
Insert a space between adjacent sets of parentheses.
Insert a + next to “112” if that number is artificially low.
Table 2.
Insert a space between the parentheses and the preceding text.
“(this paper)” is unnecessary; delete.

Supplemental material
I recommend that the analysis of placing the Merchantville specimen within theropod (as a test of the Ornithomimosaurian hypothesis) be done using the latest version of the TWiG data matrix, not the Loewen et al. (2013) data set.

---

## Round 0.4 · Minor Revisions

· Academic Editor

Minor Revisions

The manuscript is nearly ready to go, but needs a number of minor copy edits and consideration of other small issues at this point. This includes:

1) There are a number of random carriage returns throughout the manuscript; what looks like the end of a line is actually indicated as a new paragraph. Please clean this up, to make the layout process during production easier and more efficient.

2) Please include the phylogenetic matrix files as .tnt files in addition to the PDF. Please make sure to reference all supplemental info files directly in the manuscript.

3) You only need to report strict consensus trees; 50% majority rule trees leave out equally parsimonious topologies, and in any case resolution is sufficient that you do not need to worry about them.

4) I have indicated a number of minor grammatical/stylistic edits throughout on the attached PDF.

Once these issues are addressed, I should be able to approve the manuscript in short order.

---

## Round 0.5 · accepted · Accept

· Academic Editor

Accept

Thank you for your close attention to the last round of comments. Everything looks good now.